# RETHINKING LLM-BASED RAG FROM A DECOUPLED PERSPECTIVE

## ABSTRACT

This paper aims to investigate a fundamental question in LLM-based RAG (Retrieval-augmented Generation): what is the key bottleneck limiting the performance improvement of current RAG systems. This paper thereby proposes a decoupled perspective to separately analyze the potentials in retrieval and generation stages. Specifically, we design a simple method to approximating the effects of the oracle metric in retrieval stage and the oracle way to utilizing the retrieved documents in generation stage in RAG. On six classic question-answering benchmark tasks, by comparing the performance of standard RAG and its oracle variants, we observe several valuable findings: First, even with the oracle retrieval, the improvement they bring to RAG performance is not as significant as expected. Second, figuring out how to enable generation models to make good use of the retrieved documents holds greater potential for boosting RAG.

## 1 INTRODUCTION

Currently, Retrieval-Augmented Generation (RAG) (Lewis et al., 2020) serves as a core paradigm for alleviating the hallucination (Roller et al., 2021) problem of large language models (LLMs) (Brown et al., 2020). This paradigm comprises two key steps: retrieval and generation. First, it retrieves documents from an external knowledge base(**?**) that are most relevant to the query; second, it leverages these retrieved results to enable the LLM to generate more accurate responses. To improve the overall performance of RAG systems, researchers have explored various optimization paths, including the introduction of vector semantic retrieval (Karpukhin et al., 2020) and hybrid retrieval mechanisms (Sawarkar et al., 2024), re-ranking of retrieval results (Yu et al., 2024), and integration of structured knowledge sources (such as knowledge graphs) (Edge et al., 2024) to enhance the accuracy of retrieval and question-answering (Borgeaud et al., 2022). While these efforts have improved RAG performance to varying degrees, a fundamental question remains unsystematically addressed: what is the key bottleneck limiting the performance improvement of current RAG systems—does it stem from the retrieval module or the generation module? In other words, between the two core dimensions—"enhancing the quality of retrieved documents" in the retrieval stage and "enabling the LLM to better utilize retrieved documents" in the generation stage—which one offers greater potential for performance improvement and broader room for development?

To answer this question, this paper proposes a decoupled perspective to quantitatively analyze the potentials in retrieval and generation stages. This essentially requires to measure the effects of the oracle retrieval and the oracle way to utilize retrieved documents for prompting LLMs in RAG. Unfortunately, it is challenging to achieve the oracle retrieval metric and the oracle way to utilize retrieved documents. To this end, we first develop a rigorous simulation strategy to approximate both oracle effects via two simple simulation methods. Subsequently, we conduct systematic experiments on six widely used knowledge-based QA benchmark datasets with six mainstream LLMs, comprehensively assessing the actual gains of the two optimization pathways. Finally, by comparing their performances, we derive a valuable conclusion that making good use of the retrieved documents is more promising than improving the retrieval metric in RAG.

The main contributions of this paper can be summarized as follows:

- We propose, for the first time, a decoupled perspective to investigate the potentials of retrieval and generation for RAG. This is achieved by two simple simulated methods to approximate the effects of the oracle metric in retrieval and generation stages.

- Through large-scale empirical research, we draw a key empirical conclusion: even under the ideal condition where the quality of retrieved documents reaches an optimal level, the marginal improvement in overall system performance remains relatively limited, and making good use of retrieved documents is promising to improve RAG.

## 2 QUANTIFYING THE POTENTIAL OF RAG FROM A DECOUPLED PERSPECTIVE

RAG includes two stages: it firstly retrieve some relevant documents from a knowledge base in retrieval stage and then it generates the response by utilizing the retrieved documents to prompt the generator such as LLM. Hence, the entire performance of RAG depends on the quality of the retrieved documents and how to utilize the retrieved documents as the prompt. Generally, the higher the quality of the retrieved documents, the better generation performance the RAG system achieves; and a good utilization method can help the generator deal with the informative knowledge in the retrieved documents.

### 2.1 DECOUPLED PERSPECTIVE

In this section, we aim to quantify the potential of RAG from a decoupled perspective. In other words, we propose to quantify the potentials of the retrieval and generation stages, from which we can see the bottleneck in RAG as well as future directions to optimize RAG. Specifically, we mainly discuss two questions in both retrieval and generation stages:

- **Oracle metric in retrieval stage**: How to obtain the retrieved documents via the oracle metric which leads to the best RAG performance?

- **Oracle prompt in generation stage**: How to design the oracle prompt based on the retrieved documents which leads to the best RAG performance?

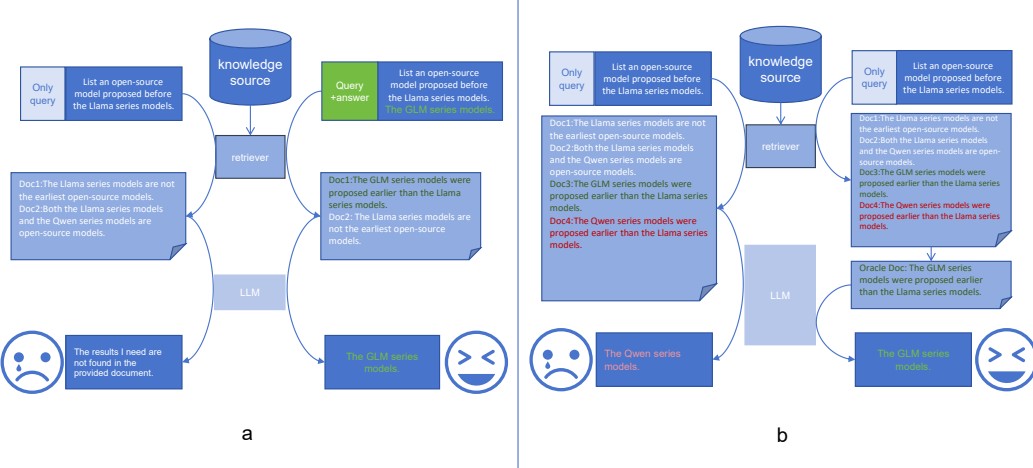

Figure 1: RAG with simulated oracle metric in retrieval stage and simulated oracle prompt in generation stage.

In the standard RAG paradigm, the retrieved documents are organized sequentially into a retrieved content according to their retrieval scores and then the standard prompt in generation is the concatenation of the query and the retrieved content. Once we obtain the oracle retrieval metric and oracle prompt, we calculate the performance gaps among the standard RAG, RAG with the oracle metric

and RAG with the oracle prompt, and we can conclude that the potential of RAG lies in the retrieval stage or utilizing the retrieved documents to prompt LLM. For example, if RAG with the oracle metric is much worse than RAG with the oracle prompt, the potential of RAG would lie in how to make good use of the retrieved documents for prompting instead of optimizing the retrieval model for RAG.

Unfortunately, it is intractable to obtain the oracle metric in retrieval stage and the oracle prompt in generation stage. Therefore, we propose two simple approximate methods to simulate the effects of the oracle metric and the oracle prompt in the rest of this section.

## 2.2 SIMULATING THE EFFECT OF ORACLE METRIC IN RETRIEVAL

To reach effect of the oracle retrieval in the RAG system, we believe that under the condition that the retriever and the retrieval knowledge base are fixed, the upper bound of retrieved content optimization is mainly related to the retrieval query. A good query can often yield more relevant documents. Therefore, we simulate this upper bound by merging the original query with the answer to form a new query. Figure 1a shows our method.

Formally, for a certain question-answer pair (question, answer), the retriever $R$ uses the question and answer as the query $Q$ to retrieve relevant documents $\mathcal{D}$ from the knowledge base $\mathcal{K}$. $\mathcal{D}$ includes multiple document segments sorted by relevance: $\{d_1, d_2, ..., d_{\text{top}k}\}$. These relevant documents, together with the question [1], are concatenated in a certain order as the prompt to the model $M$. Then, the probability that the model infers the correct answer for this question-answer pair is:

$$P_{\text{correct}} = P(M(\text{concat}(\mathcal{D}, \text{question})) = \text{answer}) \tag{1}$$

$$\mathcal{D} = \{d_1, d_2, ..., d_{\text{top}k}\} = R(\mathcal{K}, Q) \tag{2}$$

Where $P_{\text{correct}}$ denotes the probability that the model answers correctly using the retrieved documents, then "top$k$" refers to the top $k$ document segments sorted by relevance, and "cat" denotes the concatenation of texts. The modification we make to simulate the oracle retrieval is actually merging the answer into the query for retrieval, i.e.:

$$Q = \text{cat}(\text{question}, \text{answer}) \tag{3}$$

In fact, considering the actual situation, we control the probability of using the merged original query and answer as the new query to be 80% to make the effect or this oracle retrieval more achievable. In addition, for cases where there are multiple answers, to ensure convenience and fairness, we uniformly select the first one in the answer list as the standard answer.

## 2.3 SIMULATING THE EFFECT OF ORACLE PROMPT IN GENERATION

To enable more efficient utilization of retrieved documents, we hypothesize that when a sufficient number of retrieved documents are obtained, we often only need to use a small part of the key information to complete the task, while the remaining document content is often redundant, invalid, or even harmful. Therefore, we split multiple documents into independent documents and prompt the large language model to perform question-answering separately for each document. If any one of the documents can guide the model to answer correctly, we consider that there exists key information in this series of documents that can complete the task. We use this method to simulate the effect of the oracle prompt in utilizing retrieved documents.Figure 1b shows our idea. Similarly, compared with the paradigm of the naive retrieval-augmented generation system mentioned, $Q$ is equal to the question, but the probability that the model infers the correct answer for this question-answer pair is:

$$P_{\text{correct}} = 1 - \prod_{i=1}^{\text{top }k} (1 - P(M(d_i) = \text{answer})) \tag{4}$$

Here, $P_{\text{correct}}$ denotes the probability that the model answers correctly using all retrieved documents, and $d_i$ represents the retrieved sub-document utilized by the model.The probability that the model

---

[1]Some LLMs may need a system prompt besides the retrieved documents and the question, we skip the system prompt for notational simplicity.

answers correctly using all retrieved documents in this Method is equal to 1 minus the product of the probabilities that it answers incorrectly for each sub-document. It is easy to see from Equation 4 that, with other conditions fixed, the probability of the model answering the given question correctly increases as the top-k value rises.

# 3 EXPERIMENTS AND MAIN RESULTS

## 3.1 EXPERIMENTAL SETTINGS

**Datasets.** We use six classic question-answering datasets as benchmarks. For each dataset, we only select 500 samples from the test set (if there is no test set, we use the validation set; if the number of samples in the set is less than 500, we use the entire dataset directly). The datasets include 2Wiki (Ho et al., 2020), Bamboogle (Press et al., 2023), HotpotQA (Yang et al., 2018), Musique (Trivedi et al., 2022), NQ (Kwiatkowski et al., 2019), and TriviaQA (Joshi et al., 2017).The knowledge base we use is composed of open-source and downloadable Wikipedia data. We merge contents with the same title and randomly select 1,500,000 pieces of data as our knowledge base, where each piece of data includes an ID, a title, and content.

**Retrievers.** We mainly use the BM25 retrieval method to recall the top 20 retrieved documents. To conduct more in-depth analysis and obtain more convincing results, we also include the results of models using other retrievers. These retrievers include BGE-Large-EN-V1.5 (Xiao et al., 2024) and Contriever. After BM25 retrieves 1,000 documents, we calculate the similarity between the query and these retrieved documents, and then select the top 20 results by ranking as the final retrieved results.

**Models.** The models we use include the following six open-source models: DeepSeek-R1-Distill-Qwen-7B, DeepSeek-R1-0528-Qwen3-8B, Qwen2.5-7B-Instruct, Qwen3-8B, ChatGLM-9B-Chat, and Llama-3.1-8B-Instruct, all of which are mainstream open-source models.

For other experimental settings, please refer to the appendix.

## 3.2 MAIN RESULTS

**Performance Gap of RAG Before and After Optimizing Retrieved Documents**  Our experimental results include the results of direct model inference and inference with prompts using retrieved documents. We set the number of retrieved documents to five standards: 1, 2, 5, 10, and 20, and present the best results in the table 1.

It can be seen from the Table 1 that the average accuracy improvement of each model on the six datasets when using the RAG system before and after optimization ranges from 9.1% to 13.4%. This result indicates that there is a significant gap in the performance of the naive RAG system when using retrieved documents before and after optimization. However, it still cannot help the model reach a level of at least 50% accuracy in most cases, which also shows that the performance improvement brought by this optimization is limited.

**Performance Gap of RAG Before and After Optimizing the Model's Utilization of Retrieved Documents for Generation**  Our experimental results include the results of direct model inference and inference with prompts using retrieved documents. We set the number of retrieved documents to five standards: 1, 2, 5, 10, and 20, and present the best results in the Table 1. The experimental results show that the upper bound to be achieved this time is much higher. We calculated the average performance improvement of each model on different datasets, which ranges from 15.0% to 26.3%. This is a significant performance improvement, and it also indicates that even if we use ordinary queries and obtain average-quality retrieved documents, the model can make better use of the retrieved documents and achieve considerable improvement if we can accurately capture the key information.

Looking back and examining the entire Table 1, an obvious conclusion can be drawn: the performance gains from reach oracle retrieval are significantly less than those from oracle prompt in generation. In that case, in a complete RAG system, which is more important—an oracle retrieval

Table 1: Experimental results of the two optimization upper bounds across six datasets and six models. Here, "direct" denotes direct inference, "rag" represents the unoptimized RAG system, "rag (q+a)" refers to the method mentioned in Section 2.2, and "rag (s+g)" refers to the method described in Section 2.3. The numbers in the table indicate the accuracy performance of the models on the corresponding datasets.

| Model | Method | Dataset | | | | | | |
|---|---|---|---|---|---|---|---|---|
| | | 2wiki | bamboogle | hotpotqa | musique | nq | triviaqa | average |
| deepseek -r1-distill -qwen-7b | direct | 0.214 | 0.112 | 0.098 | 0.020 | 0.068 | 0.166 | 0.113 |
| | rag | 0.220 | 0.136 | 0.206 | 0.036 | 0.110 | 0.298 | 0.168 |
| | rag(q+a) | 0.238 | 0.240 | 0.296 | **0.156** | 0.232 | 0.402 | 0.261 |
| | rag(s+g) | **0.536** | **0.320** | **0.412** | 0.130 | **0.262** | **0.580** | **0.373** |
| qwen2.5 -7b -instruct | direct | 0.268 | 0.312 | 0.222 | 0.086 | 0.256 | 0.448 | 0.265 |
| | rag | 0.214 | 0.288 | 0.282 | 0.102 | 0.220 | 0.448 | 0.259 |
| | rag(q+a) | 0.294 | 0.432 | 0.360 | **0.228** | 0.366 | 0.550 | 0.372 |
| | rag(s+g) | **0.368** | **0.504** | **0.452** | 0.196 | **0.398** | **0.652** | **0.428** |
| deepseek -r1-0528 -qwen3-8b | direct | 0.272 | 0.280 | 0.250 | 0.080 | 0.220 | 0.436 | 0.256 |
| | rag | 0.210 | 0.280 | 0.286 | 0.106 | 0.204 | 0.464 | 0.258 |
| | rag(q+a) | 0.292 | 0.392 | 0.392 | **0.258** | 0.358 | 0.538 | 0.372 |
| | rag(s+g) | **0.506** | **0.544** | **0.462** | 0.210 | **0.386** | **0.704** | **0.469** |
| qwen3-8b | direct | 0.308 | 0.392 | 0.254 | 0.088 | 0.268 | 0.500 | 0.302 |
| | rag | 0.212 | 0.240 | 0.322 | 0.116 | 0.184 | 0.502 | 0.263 |
| | rag(q+a) | 0.288 | 0.416 | 0.418 | **0.270** | **0.382** | 0.604 | 0.396 |
| | rag(s+g) | **0.422** | **0.456** | **0.452** | 0.186 | 0.286 | **0.672** | **0.412** |
| chatglm -9b-chat | direct | 0.252 | 0.336 | 0.230 | 0.120 | 0.284 | 0.480 | 0.284 |
| | rag | 0.152 | 0.248 | 0.244 | 0.094 | 0.144 | 0.404 | 0.214 |
| | rag(q+a) | 0.166 | 0.336 | 0.346 | 0.186 | **0.344** | 0.516 | 0.316 |
| | rag(s+g) | **0.394** | **0.504** | **0.398** | **0.204** | 0.316 | **0.604** | **0.403** |
| llama-3.1 -8b -instruct | direct | 0.158 | 0.328 | 0.230 | 0.076 | 0.300 | 0.428 | 0.253 |
| | rag | 0.152 | 0.224 | 0.244 | 0.080 | 0.216 | 0.380 | 0.216 |
| | rag(q+a) | 0.200 | 0.312 | 0.336 | 0.186 | 0.336 | 0.470 | 0.307 |
| | rag(s+g) | 0.**442** | **0.608** | **0.478** | **0.248** | **0.550** | **0.546** | **0.479** |

or an oracle prompt in generation? In other words, which gaps can be compensated for and which are difficult to compensate for?

**Competitive Relationship Between the Benefits of oracle prompt in generation and oracle retrieval** After excluding other optimization items from the RAG system, the final choice is to reach oracle retrieval or oracle prompt in generation. In that case, the only factor to consider here is the number of retrieval-related documents. We compared the benefits of the two optimization directions based on the number of retrieved relevant documents. We compared the performance gaps of the two directions on the six datasets: if the gap difference is more than 2%, the better one gets 1 point, and the worse one gets 0 points; if the gap difference is less than 2%, it is a tie, and each gets 0.5 points. This is used to describe the performance of the better and worse directions under different numbers of documents $doc_num$. As shown in the Figure 2,

It can be seen from the figure that the intersection points corresponding to the number of retrieved documents for the two optimization upper bounds of different models are all below 10. This indicates that when we retrieve a sufficient number of texts, the benefits of oracle retrieval are no longer advantageous compared with oracle prompt in generation. At this point, the system has fallen into a bottleneck, and the model should focus more on the oracle prompt in generation.

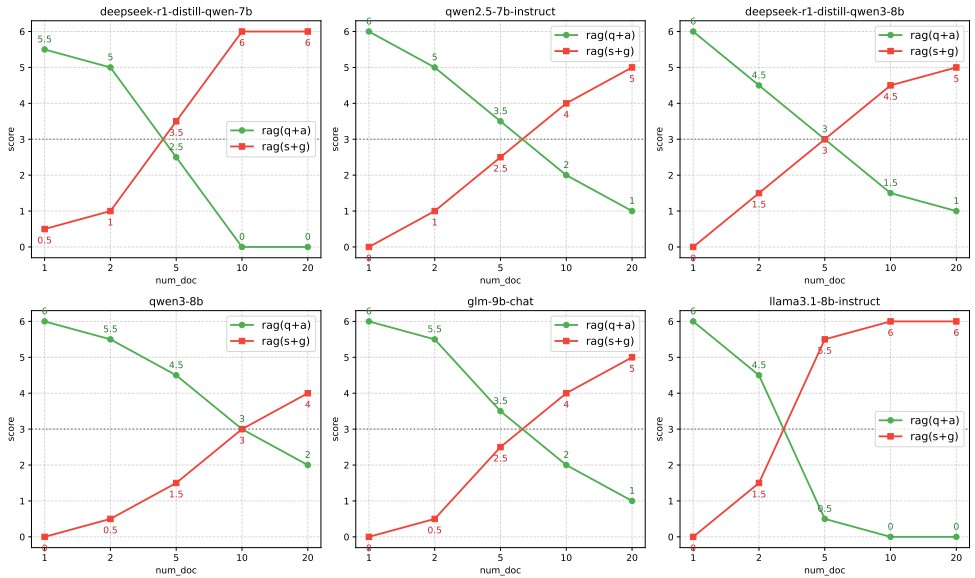

Figure 2: Relationship Between Win-Loss Scores of Performance Upper Bounds (from Two Optimization Methods) and the Number of Retrieved Documents Across Different Models.

## 4 FURTHER ANALYSIS

### 4.1 PERFORMANCE GAP CAUSED BY THE QUALITY OF RETRIEVED DOCUMENT CONTENT

**Reasons for the Limited Improvement from oracle retrieval.** We want to know whether our queries can always recall high-quality documents. Therefore, we count the proportion of dataset samples that can recall different numbers of valid documents (herein defined as documents containing answers) in different datasets. We set six ranges for the proportion of documents containing answers among the 20 retrieved documents triggered by a dataset sample, which are 0 (0%), 1 (5%), 2 (10%), 3-5 (15%-25%), 6-10 (30%-50%), and 11-20 (55%-100%). As shown in the Table 2, we can clearly see that the queries generated by some samples in certain datasets cannot find any usable knowledge documents at all. This indicates that the knowledge base fails to function to a certain extent, which is understandable in fact, because we cannot guarantee that the knowledge base can always meet complex queries. Therefore, even if we optimize the quality of retrieved documents in any way, the potential for improvement is very limited.

Table 2: The proportional distribution of each data entry across different datasets, categorized by the number of answers contained in its retrieved documents

| Percent / Dataset | 0 (0%) | 1 (5%) | 2 (10%) | 3∼5 (15∼25%) | 6∼10 (30∼50%) | 11∼20 (55∼100%) |
|---|---|---|---|---|---|---|
| 2wiki | 0.434 | 0.156 | 0.08 | 0.12 | 0.104 | 0.106 |
| Bamboogle | 0.344 | 0.12 | 0.056 | 0.12 | 0.128 | 0.232 |
| Hotpotqa | 0.22 | 0.15 | 0.102 | 0.134 | 0.164 | 0.23 |
| Musique | 0.25 | 0.16 | 0.088 | 0.174 | 0.174 | 0.154 |
| Nq | 0.204 | 0.146 | 0.048 | 0.158 | 0.166 | 0.278 |
| Triviaqa | 0.692 | 0.078 | 0.042 | 0.056 | 0.042 | 0.09 |

**Changes in Retrieved Documents.** We separately calculated the proportion of newly retrieved documents across each dataset before and after reach oracle retrieval, and attempted to identify the relationship between this proportion and the performance improvement achieved by the model on each dataset (also measured before and after optimization). Results from multiple retrievers in the Figure 3 indicate that a higher proportion of newly retrieved documents corresponds to greater

performance improvement. This confirms that the new documents obtained by adjusting the input query—compared to the old documents they replace—indeed contribute to enhancing model performance.

**Relationship with the Number of Retrieved Documents.** We actually set five standards for the number of retrieved documents (1, 2, 5, 10, and 20) to perform retrieval and return the accuracy, and counted the specific performance of the six models using retrieved documents before and after optimization. If we only focus on the performance gap caused by retrieved documents before and after optimization, the results shown in Figure 4 c indicate that a larger number of documents cannot deterministically compensate for the performance gap caused by the oracle retrieval. In other words, the performance gap here is independent of the number of documents. Therefore, the attempt to improve performance by increasing the number of documents is subject to more restrictions, and extremely long context also poses a challenge to the model's ability to understand long texts. It is worth mentioning that Figures 4 a and b show that long texts themselves cannot bring consistently better results.

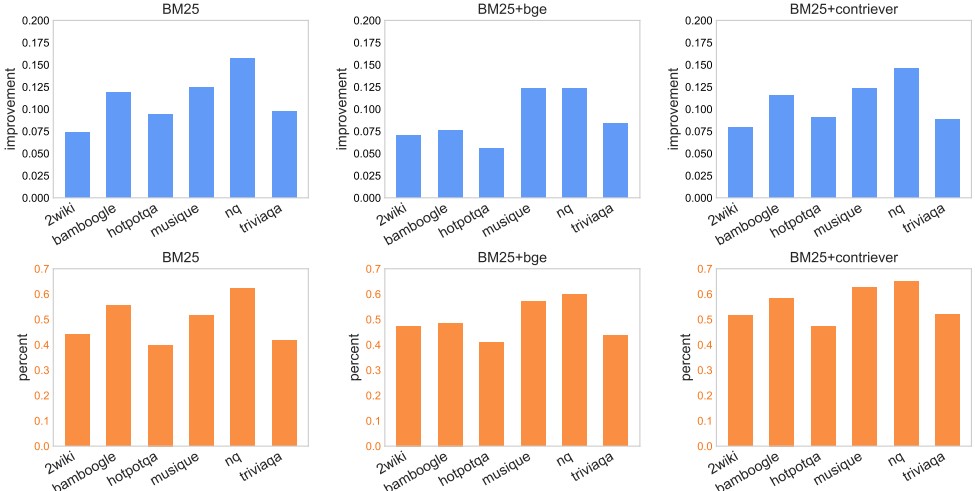

Figure 3: Average performance improvement before and after retrieved document optimization (top) and proportion of new retrieved documents before and after optimization (bottom).

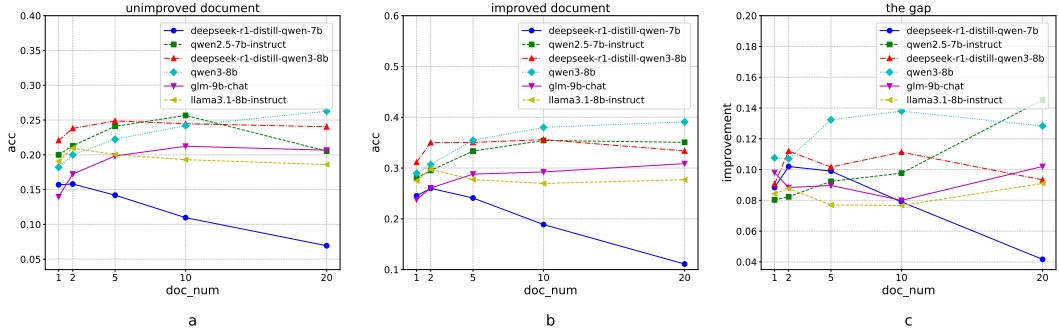

Figure 4: Accuracy of the model as a function of the number of retrieved texts before (a) and after (b) retrieved document optimization, and changes in their gap (c).

**Relationship with Retrievers.** For the sake of rigor, we tested the results of two other retrievers on the Qwen2.5-7B-Instruct and Qwen3-8B models, and also present the relationship between the performance improvement difference (after optimization) and the number of retrieved documents under the influence of different retrievers. As shown in the Figure 5, it can be found that after re-

placing the retriever, the number of retrieved documents still cannot compensate for the performance gap caused by this optimization. However, it is worth noting that replacing with a more excellent retriever can significantly reduce this performance gap. This indicates that if our retriever is more excellent, the quality of retrieved documents is more likely to be guaranteed, and thus the benefit of oracle retrieval to improve the quality of retrieved documents will be further reduced.

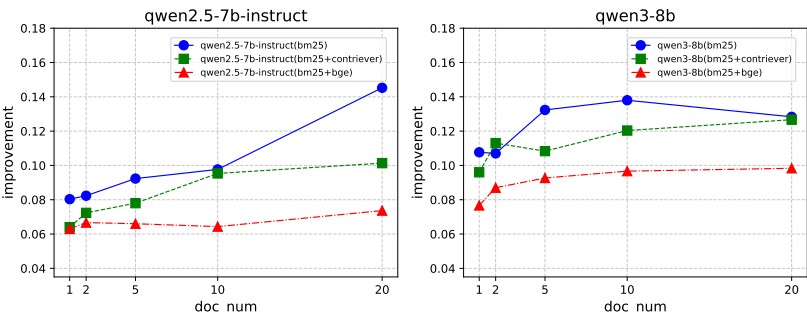

Figure 5: Changes in the gap between different retrievers before and after retrieved document optimization, as a function of the number of retrieved texts.

## 4.2 PERFORMANCE GAP CAUSED BY THE QUALITY OF RETRIEVED DOCUMENT UTILIZATION METHODS

**Relationship with the Number of Retrieved Documents.** We counted the improvement level of the simulated retrieved document utilization method under different numbers of documents $doc\_num$. It can be clearly seen from Figures 6 b and c that as the number of retrieved documents increases, the gap between before and after optimization becomes larger. This indicates that if oracle prompt in generation can be used, we can obtain more and more improvements by expanding the number of retrieved documents, although the degree of improvement gradually decreases. At the same time, such results indicate that blindly concatenating retrieved document content to help the large language model answer questions is far less accurate than extracting the most useful part. This also indirectly reflects that the large language model will be interfered by more irrelevant or incorrect information in the retrieved documents, leading to incorrect answers.

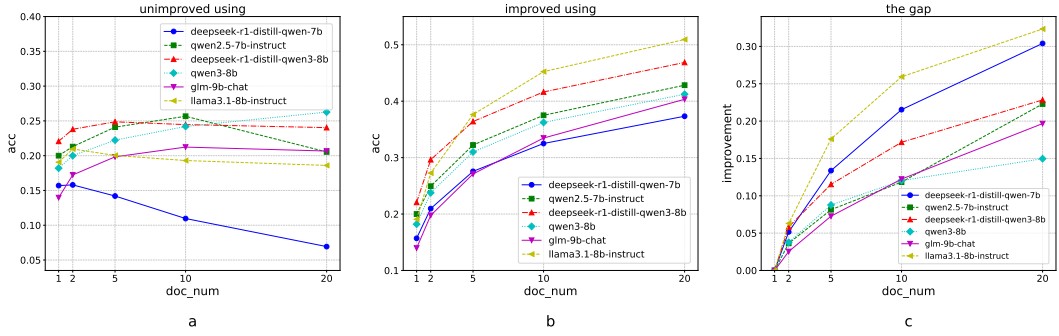

Figure 6: Accuracy of the model as a function of the number of retrieved texts before (a) and after (b) retrieved text utilization methods optimization, and changes in their gap (c).

**Stability of the Upper Bound of Document Utilization-Guided Generation.** Here, we consider the benefits brought by the upper bound of document utilization-guided generation and how much this upper bound will decrease if additional constraints are added. For example, when we require the model to use two documents simultaneously to be considered as answering correctly, we calculate how much the performance decreases compared with the case where using one document is considered correct. In the Table 3,we counted the results of the Qwen2.5-7B-Instruct and Qwen3-8B

models under this scenario: This indicates that a large part of the results obtained from the retrieved

Table 3: Variation in document utilization performance upper bounds of two models under strict constraints

| Dataset \ Model | 2wiki | Bamboogle | Hotpotqa | Musique | Nq | Triviaqa |
|---|---|---|---|---|---|---|
| qwen2.5-7b -instruct | 0.232 (-0.136) | 0.416 (-0.088) | 0.274 (-0.178) | 0.116 (-0.08) | 0.314 (-0.084) | 0.54 (-0.112) |
| qwen3-8b | 0.262 (-0.16) | 0.368 (-0.088) | 0.284 (-0.168) | 0.114 (-0.072) | 0.218 (-0.068) | 0.546 (-0.126) |

documents comes from a very specific document. Therefore, to reach the upper bound of utilization method optimization, we must accurately identify the most useful document. Similarly, based on the previous conclusions, we can also understand that although more retrieved documents increase the probability of utilizing useful documents, they also pose a challenge to the ability to find useful documents.

**Relationship with Retrievers.** Similarly, we analyzed the relationship between the performance gap of the RAG system (before and after optimizing the document utilization method) and the use of different retrievers on the Qwen2.5-7B-Instruct and Qwen3-8B models. As shown in the Figure 7, it can be seen that regardless of which retriever is used, their performance improvement gaps are almost the same. This indicates that the retriever does not significantly affect the performance gap caused by the model's retrieved document utilization method. In fact, from the previous conclusions, we can infer that a more excellent retriever can bring better retrieved documents, but the quality of document utilization is irrelevant to the quality of retrieved documents. In other words, even for low-quality retrieved documents, a sufficiently good utilization method can bring stable performance benefits.

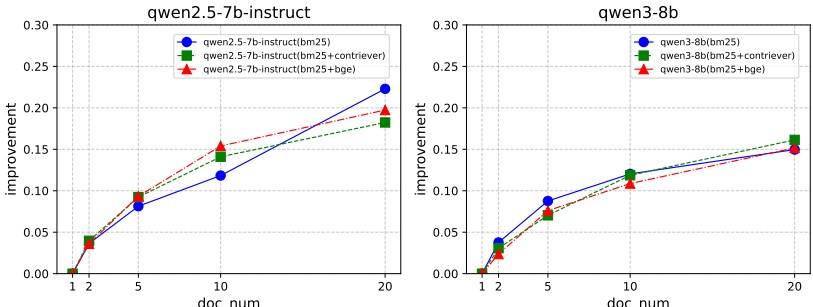

Figure 7: Changes in the gap between different retrievers before and after retrieved text utilization methods optimization, as a function of the number of retrieved texts.

## 5 CONCLUSIONS

In this work, we discussed whether the classic RAG should focus on the excellent performance of retrieved texts in terms of metrics or on how the model utilizes the documents obtained through retrieval. We drew an empirical conclusion: the bottleneck of the retrieval-augmented generation system lies in the failure to enable the model to make better use of the documents obtained through retrieval for generation. Therefore, if future work can further optimize the alignment between the long texts obtained through retrieval and the documents required for large language model question-answering, and make the model's utilization of retrieved documents more reasonable, it will be expected to break through the bottleneck of existing retrieval-augmented generation systems.

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

## A RELATED WORKS

### A.1 RETRIEVER

The earliest retrievers were dominated by sparse vector retrieval. As a word-level retrieval paradigm, it primarily employs TF-IDF to rank documents according to their relevance. Among them, the BM25 retriever (Robertson et al., 2009) is the most representative and remains a strong benchmark for many modern retrieval systems to this day. Its advantage lies in its effective keyword matching, which played a pivotal role in early-stage retrieval tasks.

Dense vector retrieval is a more modern information retrieval method that embeds queries and documents into a continuous vector space. Such retrievers are often transformer-based models (Vaswani et al., 2017) trained on large amounts of data and directly used as embedding models for retrieval (e.g., BERT (Devlin et al., 2019)). They adopt a dual-encoding architecture, encoding queries and documents separately, and achieve efficient retrieval based on vector similarity calculation. Compared with sparse vector retrieval, dense vector retrieval can better capture semantic relationships, even for sources in different languages. Moreover, with the improvement of model performance, the capabilities of retrievers have also been enhanced, leading to the emergence of more powerful retrievers such as Contriever (Izacard et al.) and BGE-M3 (Chen et al., 2024).

Hybrid retrieval combines sparse vector retrieval and dense vector retrieval, and is an effective method that can focus on both the central theme of text segments and global features. Feng et al. (2024) have proposed first defining the knowledge domain required for a query as a fixed professional domain, and then using dense retrieval within this domain to recall supplementary information. This retrieval method not only ensures that the retrieved content is relevant to the domain but also guarantees that the retrieved documents are close to the semantics of the query.

### A.2 GENERATION MODEL

The origin of existing generation models can be attributed to the establishment of the transformer architecture in the early days. The pioneering attention mechanism has changed the structural paradigm of all subsequent generation models. GPT-2 (Radford et al., 2019) is regarded as the starting point of early generative pre-trained models. Later, this series of models underwent pre-training and fine-tuning with larger datasets and more parameters, leading to the release of the closed-source GPT-3 model (Brown et al., 2020). Such closed-source models can achieve quite excellent performance on downstream tasks without additional training, which also promotes the further development of models towards ultra-large-scale language models.

In recent years, open-source large language models have achieved rapid progress both in quantity and performance, including the well-known Llama (Touvron et al., 2023a;b; Dubey et al., 2024), Qwen (Bai et al., 2023; Team, 2024), and GLM (Du et al., 2022) series. These models have achieved quite outstanding performance in knowledge question-answering, instruction following, mathematical capabilities, and code writing. Among them, the DeepSeek-R1 model (Liu et al., 2024), as a reasoning model, has achieved performance comparable to that of the then most powerful closed-source GPT-4 (Achiam et al., 2023) series through training methods based on reinforcement learning. This further reveals the importance of reinforcement learning in the training of large language generation models, leading to the emergence of a large number of reasoning models that output intermediate thinking processes and possess quite strong reasoning capabilities.

Silver & Sutton (2025) have shown that the existing training data has approached the boundary of human knowledge. Therefore, the focus of current model training has shifted from large-scale pre-training and fine-tuning to methods such as reinforcement learning and distillation, and the scale expansion of generation models has gradually ceased. These models meet the needs of daily development and lay the foundation for the construction of various LLM systems.

### A.3 RETRIEVAL-AUGMENTED GENERATION SYSTEM

A Retrieval-Augmented Generation system is a hybrid architecture that includes two key components: a retrieval system and a generation system. RAG aims to address the respective limitations of

retrieval systems and generation systems. Retrieval systems lack the ability to integrate documents for accurate and organized description, while generation systems often produce inaccurate results.

Early attempts by researchers can be traced back to studies such as DrQA (Chen et al., 2017), which used retrieval technology to obtain relevant documents for question-answering tasks, but the role of the generation model was negligible. It was not until studies like REALM (Moniz et al., 2024) proposed the alignment of retrieval and generation components that the basic paradigm of RAG was basically established.

Subsequent researchers have carried out a great deal of optimization work on RAG: in terms of performance, optimization measures include decomposing retrieval queries (Song & Zheng, 2024) to make retrieved documents more accurate, re-ranking (Yu et al., 2024), filtering (Yan et al., 2024), and screening the retrieved documents for model training, and optimizing the reasoning process (Wang et al., 2024) of the generation model for question-answering; the combination of RAG systems with specific domain tasks (such as medical-related tasks (Wu et al., 2024)) has also achieved considerable performance improvement; at the same time, the RAG field can be extended to multi-modal information (Alayrac et al., 2022), becoming a key direction for optimizing the multi-modal question-answering capabilities of models. Nowadays, Deep Research (Zheng et al., 2025), as a more advanced system based on the RAG system, can handle more complex and challenging application scenarios.

# B USE OF LLMS

We utilized a variety of large language models (LLMs) to assist in polishing our paper and correcting flaws, with the goal of presenting readers with a more idiomatic English academic manuscript. To achieve this, we conducted a brief comparison of different LLMs and the use of various prompts. Additionally, when handling the extensive table content in the appendix, we adopted LaTeX outputs generated by LLMs efficiently, rather than performing repetitive manual operations.

# C SUPPLEMENTARY EXPERIMENTAL MATERIALS

## C.1 PROMPT TEMPLATES USED FOR THE MODEL

1. **Direct Inference**

```
Please answer the following question. You should think step
by step to solve it.
Provide your final answer in the format \boxed{YOUR_ANSWER}.
Question:
{question}
```

2. **RAG**

```
Please answer the following question. You should think step
by step to solve it.
You are a knowledgeable assistant that uses the provided
documents to answer the user's question.
Question:
{question}
Documents:
{documents}
```

## C.2 APPROXIMATE RELATIONSHIP BETWEEN THE NUMBER OF RETRIEVED DOCUMENTS AND THEIR AVERAGE TOKEN CONSUMPTION

This section presents the number of tokens that the model needs to process, corresponding to different quantities of retrieved documents. In our experiments, the maximum length of each document is

uniformly set to 30k tokens—this value falls within the context length range supported by all participating models. As shown in the table 4 5 below, 30k tokens is a reasonable setting that exceeds the average context length level.

Table 4: Models and their corresponding maximum context lengths

| Model | Max_len |
|---|---|
| deepseek-r1-distill-qwen-7b | 131072 |
| qwen2.5-7b-instruct | 32768 |
| deepseek-r1-0528-qwen3-8b | 131072 |
| qwen3-8b | 40960 |
| chatglm-9b-chat | 131072 |
| llama-3.1-8b-instruct | 131072 |

Table 5: Token consumption counts corresponding to each dataset

| Doc_num
Dataset | 1 | 2 | 5 | 10 | 20 |
|---|---|---|---|---|---|
| 2wiki | 841 | 1757 | 4488 | 9132 | 18576 |
| Bamboogle | 1266 | 2684 | 6490 | 12376 | 25516 |
| Hotpotqa | 1378 | 2690 | 6612 | 13251 | 26573 |
| Musique | 1309 | 2554 | 6540 | 13188 | 25867 |
| Nq | 1165 | 2317 | 6111 | 12138 | 24303 |
| Triviaqa | 1381 | 2699 | 6467 | 13198 | 26334 |
| Average | 1233 | 2450 | 6118 | 12213 | 24528 |

## C.3 ALL COMPLETE EXPERIMENTAL RESULTS

This section documents all experimental results of our model's inference accuracy. Here, a doc_num of 0 indicates direct inference. Consistent with the previous context, "rag" represents the standard RAG method, "rag (q+a)" denotes the simulation of reaching the upper bound mentioned in Section 4.1, and "rag (s+g)" denotes the simulation of reaching the upper bound described in Section 4.2. Each table is labeled with the format "Model,Retriever" to facilitate cross-reference and review.

Table 6: deepseek-r1-distill-qwen-7b,bm25

| Dataset | Method | num_docs | | | | | |
|---|---|---|---|---|---|---|---|
| | | 0 | 1 | 2 | 5 | 10 | 20 |
| 2wiki | rag | 0.214 | 0.220 | 0.184 | 0.152 | 0.150 | 0.150 |
| | rag(q+a) | 0.214 | 0.220 | 0.236 | 0.238 | 0.232 | 0.192 |
| | rag(s+g) | 0.214 | 0.220 | 0.270 | 0.422 | 0.496 | 0.536 |
| bamboogle | rag | 0.112 | 0.120 | 0.128 | 0.136 | 0.088 | 0.048 |
| | rag(q+a) | 0.112 | 0.232 | 0.240 | 0.208 | 0.176 | 0.080 |
| | rag(s+g) | 0.112 | 0.120 | 0.184 | 0.208 | 0.248 | 0.320 |
| hotpotqa | rag | 0.098 | 0.180 | 0.206 | 0.180 | 0.126 | 0.078 |
| | rag(q+a) | 0.098 | 0.278 | 0.296 | 0.254 | 0.200 | 0.130 |
| | rag(s+g) | 0.098 | 0.180 | 0.236 | 0.326 | 0.370 | 0.412 |
| musique | rag | 0.020 | 0.032 | 0.036 | 0.030 | 0.026 | 0.002 |
| | rag(q+a) | 0.020 | 0.138 | 0.154 | 0.156 | 0.076 | 0.048 |
| | rag(s+g) | 0.020 | 0.032 | 0.058 | 0.072 | 0.096 | 0.130 |
| nq | rag | 0.068 | 0.110 | 0.096 | 0.082 | 0.050 | 0.028 |
| | rag(q+a) | 0.068 | 0.210 | 0.232 | 0.222 | 0.162 | 0.092 |
| | rag(s+g) | 0.068 | 0.110 | 0.160 | 0.180 | 0.224 | 0.262 |
| triviaqa | rag | 0.166 | 0.280 | 0.298 | 0.272 | 0.218 | 0.110 |
| | rag(q+a) | 0.166 | 0.394 | 0.402 | 0.368 | 0.286 | 0.124 |
| | rag(s+g) | 0.166 | 0.280 | 0.350 | 0.446 | 0.516 | 0.580 |

Table 7: qwen2.5-7b-instruct,bm25

| Dataset | Method | num_docs | | | | | |
|---|---|---|---|---|---|---|---|
| | | 0 | 1 | 2 | 5 | 10 | 20 |
| 2wiki | rag | 0.268 | 0.128 | 0.142 | 0.192 | 0.200 | 0.214 |
| | rag(q+a) | 0.268 | 0.200 | 0.184 | 0.230 | 0.286 | 0.294 |
| | rag(s+g) | 0.268 | 0.128 | 0.174 | 0.232 | 0.312 | 0.368 |
| bamboogle | rag | 0.312 | 0.240 | 0.256 | 0.264 | 0.288 | 0.240 |
| | rag(q+a) | 0.312 | 0.304 | 0.304 | 0.432 | 0.352 | 0.368 |
| | rag(s+g) | 0.312 | 0.240 | 0.296 | 0.368 | 0.408 | 0.504 |
| hotpotqa | rag | 0.222 | 0.236 | 0.228 | 0.278 | 0.282 | 0.220 |
| | rag(q+a) | 0.222 | 0.310 | 0.330 | 0.346 | 0.360 | 0.358 |
| | rag(s+g) | 0.222 | 0.236 | 0.288 | 0.362 | 0.410 | 0.452 |
| musique | rag | 0.086 | 0.074 | 0.076 | 0.100 | 0.102 | 0.075 |
| | rag(q+a) | 0.086 | 0.156 | 0.168 | 0.194 | 0.212 | 0.228 |
| | rag(s+g) | 0.086 | 0.074 | 0.096 | 0.148 | 0.164 | 0.196 |
| nq | rag | 0.256 | 0.158 | 0.172 | 0.198 | 0.220 | 0.172 |
| | rag(q+a) | 0.256 | 0.270 | 0.294 | 0.304 | 0.366 | 0.34 |
| | rag(s+g) | 0.256 | 0.158 | 0.212 | 0.298 | 0.354 | 0.398 |
| triviaqa | rag | 0.448 | 0.364 | 0.404 | 0.414 | 0.448 | 0.312 |
| | rag(q+a) | 0.448 | 0.442 | 0.492 | 0.494 | 0.550 | 0.516 |
| | rag(s+g) | 0.448 | 0.364 | 0.430 | 0.526 | 0.602 | 0.652 |

Table 8: qwen2.5-7b-instruct,bm25+contriever

| Dataset | Method | num_docs | | | | | |
|---|---|---|---|---|---|---|---|
| | | 0 | 1 | 2 | 5 | 10 | 20 |
| 2wiki | rag | 0.268 | 0.108 | 0.164 | 0.198 | 0.234 | 0.274 |
| | rag(q+a) | 0.268 | 0.140 | 0.198 | 0.234 | 0.282 | 0.346 |
| | rag(s+g) | 0.268 | 0.108 | 0.178 | 0.274 | 0.342 | 0.420 |
| bamboogle | rag | 0.312 | 0.296 | 0.312 | 0.328 | 0.264 | 0.280 |
| | rag(q+a) | 0.312 | 0.336 | 0.352 | 0.384 | 0.416 | 0.392 |
| | rag(s+g) | 0.312 | 0.296 | 0.368 | 0.432 | 0.520 | 0.560 |
| hotpotqa | rag | 0.222 | 0.188 | 0.224 | 0.242 | 0.254 | 0.282 |
| | rag(q+a) | 0.222 | 0.258 | 0.298 | 0.342 | 0.328 | 0.356 |
| | rag(s+g) | 0.222 | 0.188 | 0.252 | 0.324 | 0.370 | 0.416 |
| musique | rag | 0.086 | 0.084 | 0.088 | 0.100 | 0.126 | 0.106 |
| | rag(q+a) | 0.086 | 0.154 | 0.176 | 0.184 | 0.222 | 0.204 |
| | rag(s+g) | 0.086 | 0.084 | 0.130 | 0.164 | 0.186 | 0.216 |
| nq | rag | 0.256 | 0.208 | 0.220 | 0.226 | 0.236 | 0.206 |
| | rag(q+a) | 0.256 | 0.306 | 0.340 | 0.368 | 0.362 | 0.372 |
| | rag(s+g) | 0.256 | 0.208 | 0.270 | 0.346 | 0.390 | 0.422 |
| triviaqa | rag | 0.448 | 0.400 | 0.438 | 0.470 | 0.476 | 0.464 |
| | rag(q+a) | 0.448 | 0.474 | 0.516 | 0.520 | 0.552 | 0.550 |
| | rag(s+g) | 0.448 | 0.400 | 0.486 | 0.578 | 0.628 | 0.672 |

Table 9: qwen2.5-7b-instruct,bm25+bge

| Dataset | Method | num_docs | | | | | |
|---|---|---|---|---|---|---|---|
| | | 0 | 1 | 2 | 5 | 10 | 20 |
| 2wiki | rag | 0.268 | 0.132 | 0.204 | 0.230 | 0.248 | 0.256 |
| | rag(q+a) | 0.268 | 0.164 | 0.198 | 0.262 | 0.280 | 0.324 |
| | rag(s+g) | 0.268 | 0.132 | 0.196 | 0.308 | 0.384 | 0.464 |
| bamboogle | rag | 0.312 | 0.256 | 0.288 | 0.320 | 0.304 | 0.288 |
| | rag(q+a) | 0.312 | 0.296 | 0.368 | 0.320 | 0.304 | 0.368 |
| | rag(s+g) | 0.312 | 0.256 | 0.344 | 0.416 | 0.528 | 0.592 |
| hotpotqa | rag | 0.222 | 0.236 | 0.272 | 0.296 | 0.278 | 0.300 |
| | rag(q+a) | 0.222 | 0.284 | 0.338 | 0.350 | 0.352 | 0.340 |
| | rag(s+g) | 0.222 | 0.236 | 0.298 | 0.368 | 0.434 | 0.452 |
| musique | rag | 0.086 | 0.084 | 0.126 | 0.124 | 0.120 | 0.136 |
| | rag(q+a) | 0.086 | 0.184 | 0.198 | 0.226 | 0.210 | 0.214 |
| | rag(s+g) | 0.086 | 0.084 | 0.144 | 0.188 | 0.198 | 0.228 |
| nq | rag | 0.256 | 0.220 | 0.234 | 0.242 | 0.248 | 0.234 |
| | rag(q+a) | 0.256 | 0.310 | 0.344 | 0.360 | 0.342 | 0.338 |
| | rag(s+g) | 0.256 | 0.220 | 0.284 | 0.354 | 0.406 | 0.448 |
| triviaqa | rag | 0.448 | 0.458 | 0.476 | 0.490 | 0.488 | 0.482 |
| | rag(q+a) | 0.448 | 0.526 | 0.554 | 0.580 | 0.584 | 0.554 |
| | rag(s+g) | 0.448 | 0.458 | 0.550 | 0.630 | 0.660 | 0.696 |

Table 10: deepseek-r1-0528-qwen3-8b,bm25

| Dataset | Method | num_docs | | | | | |
|---|---|---|---|---|---|---|---|
| | | 0 | 1 | 2 | 5 | 10 | 20 |
| 2wiki | rag | 0.272 | 0.188 | 0.164 | 0.196 | 0.206 | 0.210 |
| | rag(q+a) | 0.272 | 0.230 | 0.240 | 0.240 | 0.252 | 0.292 |
| | rag(s+g) | 0.272 | 0.188 | 0.290 | 0.372 | 0.438 | 0.506 |
| bamboogle | rag | 0.280 | 0.224 | 0.264 | 0.280 | 0.216 | 0.224 |
| | rag(q+a) | 0.280 | 0.328 | 0.392 | 0.352 | 0.376 | 0.296 |
| | rag(s+g) | 0.280 | 0.224 | 0.312 | 0.384 | 0.456 | 0.544 |
| hotpotqa | rag | 0.250 | 0.228 | 0.268 | 0.284 | 0.282 | 0.286 |
| | rag(q+a) | 0.250 | 0.324 | 0.370 | 0.392 | 0.378 | 0.348 |
| | rag(s+g) | 0.250 | 0.228 | 0.296 | 0.370 | 0.420 | 0.462 |
| musique | rag | 0.080 | 0.078 | 0.090 | 0.094 | 0.102 | 0.106 |
| | rag(q+a) | 0.080 | 0.190 | 0.236 | 0.228 | 0.258 | 0.232 |
| | rag(s+g) | 0.080 | 0.078 | 0.108 | 0.150 | 0.182 | 0.210 |
| nq | rag | 0.220 | 0.172 | 0.188 | 0.188 | 0.198 | 0.204 |
| | rag(q+a) | 0.220 | 0.294 | 0.324 | 0.358 | 0.350 | 0.342 |
| | rag(s+g) | 0.220 | 0.172 | 0.244 | 0.304 | 0.346 | 0.386 |
| triviaqa | rag | 0.436 | 0.434 | 0.454 | 0.450 | 0.464 | 0.412 |
| | rag(q+a) | 0.436 | 0.502 | 0.538 | 0.532 | 0.522 | 0.492 |
| | rag(s+g) | 0.436 | 0.434 | 0.528 | 0.604 | 0.656 | 0.704 |

Table 11: qwen3-8b,bm25

| Dataset | Method | num_docs | | | | | |
|---|---|---|---|---|---|---|---|
| | | 0 | 1 | 2 | 5 | 10 | 20 |
| 2wiki | rag | 0.308 | 0.140 | 0.162 | 0.180 | 0.184 | 0.212 |
| | rag(q+a) | 0.308 | 0.204 | 0.192 | 0.238 | 0.264 | 0.288 |
| | rag(s+g) | 0.308 | 0.140 | 0.186 | 0.272 | 0.348 | 0.422 |
| bamboogle | rag | 0.392 | 0.160 | 0.224 | 0.232 | 0.240 | 0.240 |
| | rag(q+a) | 0.392 | 0.320 | 0.344 | 0.400 | 0.416 | 0.392 |
| | rag(s+g) | 0.392 | 0.160 | 0.248 | 0.344 | 0.400 | 0.456 |
| hotpotqa | rag | 0.254 | 0.218 | 0.256 | 0.272 | 0.304 | 0.322 |
| | rag(q+a) | 0.254 | 0.336 | 0.350 | 0.390 | 0.418 | 0.410 |
| | rag(s+g) | 0.254 | 0.218 | 0.294 | 0.358 | 0.402 | 0.452 |
| musique | rag | 0.088 | 0.066 | 0.058 | 0.100 | 0.104 | 0.116 |
| | rag(q+a) | 0.088 | 0.178 | 0.194 | 0.222 | 0.256 | 0.270 |
| | rag(s+g) | 0.088 | 0.066 | 0.092 | 0.134 | 0.158 | 0.186 |
| nq | rag | 0.268 | 0.124 | 0.118 | 0.124 | 0.150 | 0.184 |
| | rag(q+a) | 0.268 | 0.218 | 0.270 | 0.322 | 0.356 | 0.382 |
| | rag(s+g) | 0.268 | 0.124 | 0.158 | 0.208 | 0.244 | 0.286 |
| triviaqa | rag | 0.500 | 0.384 | 0.382 | 0.426 | 0.470 | 0.502 |
| | rag(q+a) | 0.500 | 0.482 | 0.492 | 0.556 | 0.570 | 0.604 |
| | rag(s+g) | 0.500 | 0.384 | 0.448 | 0.544 | 0.622 | 0.672 |

Table 12: qwen3-8b,bm25+contriever

| Dataset | Method | num_docs | | | | | |
|---|---|---|---|---|---|---|---|
| | | 0 | 1 | 2 | 5 | 10 | 20 |
| 2wiki | rag | 0.308 | 0.136 | 0.146 | 0.194 | 0.210 | 0.238 |
| | rag(q+a) | 0.308 | 0.190 | 0.210 | 0.246 | 0.278 | 0.326 |
| | rag(s+g) | 0.308 | 0.136 | 0.162 | 0.282 | 0.372 | 0.456 |
| bamboogle | rag | 0.392 | 0.232 | 0.256 | 0.280 | 0.232 | 0.272 |
| | rag(q+a) | 0.392 | 0.312 | 0.336 | 0.360 | 0.392 | 0.424 |
| | rag(s+g) | 0.392 | 0.232 | 0.296 | 0.360 | 0.464 | 0.504 |
| hotpotqa | rag | 0.254 | 0.216 | 0.234 | 0.262 | 0.292 | 0.288 |
| | rag(q+a) | 0.254 | 0.302 | 0.350 | 0.378 | 0.386 | 0.400 |
| | rag(s+g) | 0.254 | 0.216 | 0.254 | 0.330 | 0.392 | 0.448 |
| musique | rag | 0.088 | 0.066 | 0.076 | 0.096 | 0.112 | 0.114 |
| | rag(q+a) | 0.088 | 0.166 | 0.230 | 0.226 | 0.236 | 0.266 |
| | rag(s+g) | 0.088 | 0.066 | 0.102 | 0.134 | 0.172 | 0.212 |
| nq | rag | 0.268 | 0.172 | 0.204 | 0.224 | 0.238 | 0.252 |
| | rag(q+a) | 0.268 | 0.312 | 0.346 | 0.392 | 0.408 | 0.408 |
| | rag(s+g) | 0.268 | 0.172 | 0.234 | 0.278 | 0.324 | 0.354 |
| triviaqa | rag | 0.500 | 0.408 | 0.440 | 0.492 | 0.534 | 0.546 |
| | rag(q+a) | 0.500 | 0.524 | 0.562 | 0.596 | 0.640 | 0.646 |
| | rag(s+g) | 0.500 | 0.408 | 0.492 | 0.586 | 0.606 | 0.704 |

Table 13: qwen3-8b,bm25+bge

| Dataset | Method | num_docs | | | | | |
|---|---|---|---|---|---|---|---|
| | | 0 | 1 | 2 | 5 | 10 | 20 |
| 2wiki | rag | 0.308 | 0.148 | 0.200 | 0.222 | 0.236 | 0.258 |
| | rag(q+a) | 0.308 | 0.176 | 0.212 | 0.264 | 0.332 | 0.332 |
| | rag(s+g) | 0.308 | 0.148 | 0.222 | 0.316 | 0.400 | 0.468 |
| bamboogle | rag | 0.392 | 0.280 | 0.312 | 0.272 | 0.304 | 0.312 |
| | rag(q+a) | 0.392 | 0.312 | 0.368 | 0.384 | 0.384 | 0.416 |
| | rag(s+g) | 0.392 | 0.280 | 0.312 | 0.368 | 0.448 | 0.504 |
| hotpotqa | rag | 0.254 | 0.244 | 0.250 | 0.304 | 0.330 | 0.320 |
| | rag(q+a) | 0.254 | 0.320 | 0.344 | 0.364 | 0.380 | 0.390 |
| | rag(s+g) | 0.254 | 0.244 | 0.298 | 0.384 | 0.432 | 0.468 |
| musique | rag | 0.088 | 0.100 | 0.126 | 0.146 | 0.146 | 0.142 |
| | rag(q+a) | 0.088 | 0.192 | 0.260 | 0.254 | 0.294 | 0.274 |
| | rag(s+g) | 0.088 | 0.100 | 0.142 | 0.174 | 0.194 | 0.236 |
| nq | rag | 0.268 | 0.200 | 0.234 | 0.230 | 0.252 | 0.270 |
| | rag(q+a) | 0.268 | 0.336 | 0.370 | 0.386 | 0.394 | 0.406 |
| | rag(s+g) | 0.268 | 0.200 | 0.248 | 0.304 | 0.346 | 0.392 |
| triviaqa | rag | 0.500 | 0.480 | 0.520 | 0.542 | 0.562 | 0.568 |
| | rag(q+a) | 0.500 | 0.576 | 0.610 | 0.620 | 0.626 | 0.642 |
| | rag(s+g) | 0.500 | 0.480 | 0.562 | 0.626 | 0.662 | 0.712 |

Table 14: glm-9b-chat,bm25

| Dataset | Method | num_docs | | | | | |
|---|---|---|---|---|---|---|---|
| | | 0 | 1 | 2 | 5 | 10 | 20 |
| 2wiki | rag | 0.252 | 0.076 | 0.100 | 0.114 | 0.152 | 0.128 |
| | rag(q+a) | 0.252 | 0.110 | 0.118 | 0.136 | 0.150 | 0.166 |
| | rag(s+g) | 0.252 | 0.076 | 0.108 | 0.204 | 0.286 | 0.394 |
| bamboogle | rag | 0.336 | 0.144 | 0.216 | 0.248 | 0.240 | 0.232 |
| | rag(q+a) | 0.336 | 0.256 | 0.288 | 0.336 | 0.320 | 0.296 |
| | rag(s+g) | 0.336 | 0.144 | 0.224 | 0.328 | 0.408 | 0.504 |
| hotpotqa | rag | 0.230 | 0.170 | 0.208 | 0.226 | 0.244 | 0.240 |
| | rag(q+a) | 0.230 | 0.266 | 0.290 | 0.308 | 0.288 | 0.346 |
| | rag(s+g) | 0.230 | 0.170 | 0.236 | 0.298 | 0.346 | 0.398 |
| musique | rag | 0.120 | 0.058 | 0.060 | 0.090 | 0.090 | 0.094 |
| | rag(q+a) | 0.120 | 0.156 | 0.170 | 0.174 | 0.186 | 0.184 |
| | rag(s+g) | 0.120 | 0.058 | 0.094 | 0.136 | 0.166 | 0.204 |
| nq | rag | 0.284 | 0.092 | 0.104 | 0.120 | 0.144 | 0.144 |
| | rag(q+a) | 0.284 | 0.228 | 0.250 | 0.298 | 0.296 | 0.344 |
| | rag(s+g) | 0.284 | 0.092 | 0.138 | 0.202 | 0.270 | 0.316 |
| triviaqa | rag | 0.480 | 0.298 | 0.346 | 0.392 | 0.404 | 0.402 |
| | rag(q+a) | 0.480 | 0.410 | 0.448 | 0.476 | 0.514 | 0.516 |
| | rag(s+g) | 0.480 | 0.298 | 0.386 | 0.458 | 0.538 | 0.604 |

Table 15: llama3.1-8b-instruct,bm25

| Dataset | Method | num_docs | | | | | |
|---|---|---|---|---|---|---|---|
| | | 0 | 1 | 2 | 5 | 10 | 20 |
| 2wiki | rag | 0.158 | 0.118 | 0.138 | 0.124 | 0.152 | 0.130 |
| | rag(q+a) | 0.158 | 0.156 | 0.200 | 0.166 | 0.160 | 0.154 |
| | rag(s+g) | 0.158 | 0.118 | 0.178 | 0.266 | 0.360 | 0.442 |
| bamboogle | rag | 0.328 | 0.200 | 0.224 | 0.208 | 0.192 | 0.184 |
| | rag(q+a) | 0.328 | 0.288 | 0.304 | 0.304 | 0.280 | 0.312 |
| | rag(s+g) | 0.328 | 0.200 | 0.288 | 0.432 | 0.552 | 0.608 |
| hotpotqa | rag | 0.230 | 0.200 | 0.224 | 0.238 | 0.244 | 0.220 |
| | rag(q+a) | 0.230 | 0.312 | 0.316 | 0.306 | 0.310 | 0.336 |
| | rag(s+g) | 0.230 | 0.200 | 0.286 | 0.368 | 0.424 | 0.478 |
| musique | rag | 0.076 | 0.072 | 0.078 | 0.080 | 0.072 | 0.076 |
| | rag(q+a) | 0.076 | 0.150 | 0.186 | 0.154 | 0.144 | 0.138 |
| | rag(s+g) | 0.076 | 0.072 | 0.124 | 0.170 | 0.202 | 0.248 |
| nq | rag | 0.300 | 0.200 | 0.216 | 0.172 | 0.168 | 0.156 |
| | rag(q+a) | 0.300 | 0.274 | 0.336 | 0.316 | 0.284 | 0.294 |
| | rag(s+g) | 0.300 | 0.200 | 0.290 | 0.424 | 0.488 | 0.550 |
| triviaqa | rag | 0.428 | 0.354 | 0.378 | 0.380 | 0.330 | 0.350 |
| | rag(q+a) | 0.428 | 0.470 | 0.442 | 0.418 | 0.440 | 0.428 |
| | rag(s+g) | 0.428 | 0.354 | 0.470 | 0.598 | 0.688 | 0.730 |

