# OpenReview forum: "Rethinking LLM-based RAG from a decoupled perspective"
_ICLR.cc/2026/Conference — ICLR 2026 Conference Withdrawn Submission_

### Official Review · Reviewer_sPFa · 2025-10-24

**Soundness:** 2
**Presentation:** 2
**Contribution:** 2
**Rating:** 2
**Confidence:** 4

**Summary:**

This paper studies in RAG systems, which is the main quality bottleneck between retriever and answer generation. This is an interesting problem to study; however, the answer would highly rely on the model capability. The paper makes conclusions pre-maturely w/o analyzing diverse sets of models, question types, and retrieval corpus.

**Strengths:**

S1. The paper studies an interesting question: in RAG systems, which is the main quality bottleneck between retriever and answer generation?

S2. The underlying idea, assuming optimal solution in one step, and study effect of the other step, is intuitive.

S3. The idea of generating optimal retrieval results by search the concatenation of question and answer is smart.

**Weaknesses:**

W1. The method in Sec 2.2 assumes a "simple" question where the answer can be directly found in a document. What if a multi-hop question where you need to get use multiple documents for the multiple hops? What if an aggregation question where you need to get elementary results from multiple documents for aggregation?

W2. The method in Sec 2.3 is unclear. In particular, how to compute P(M(d_i)=answer)? Does it mean as far as we can get the correct answer from one document, P_{correct} = 1? What if the model cannot find the answer from any of the document, but can find the answer if we only show the chunk (e.g., paragraph) that gives the correct answer?

W3. The experiment datasets are limited. 1) The only retrieval corpus is Wikipedia, which does not necessarily represent all different kinds of data available on the web. 2) Latest RAG benchmark CRAG shall be considered, since it contains various types of questions and various types of retrieval corpus.

W4. All models studied are small models, 7-9B. It is questionable how strong these models are in handling retrieval noises and conducting complex reasonings. As such, the conclusions made on these models may not apply when using stronger models.

W5. The analysis considers only correctness, but didn't separate missing answers and incorrect (hallucinated) answers. Since they have different impacts on user experiences, we shall distinguish them in comparing. A solution that makes a lot of  mistakes may not be better than a solution that honestly admit "I don't know".

W6. Finally, when there are a few steps, each step has its importance. For RAG, assuming the model cannot answer the question by itself, e2e QA accuracy is the product of retrieval recall and answer generation accuracy from correct retrieval results. For different benchmarks, different retrieval corpus, different models, the two terms can differ. It is not appropriate to say "made better use of retrieved documents" is always the key bottleneck.

W7. Writing can be improved. The conclusion is not super clear; key insights are unclear.

**Questions:**

Please reply to the weaknesses.

---

### Official Review · Reviewer_1rG3 · 2025-10-30

**Soundness:** 2
**Presentation:** 2
**Contribution:** 2
**Rating:** 2
**Confidence:** 5

**Summary:**

The paper studies where the performance headroom of RAG actually lies by decoupling retrieval and generation. It simulates an “oracle retrieval” by augmenting the query with the gold answer and an “oracle utilization” by prompting the LLM with each retrieved chunk independently, then compares gaps across six QA datasets and several open-source LLMs. The headline takeaway is that gains from better utilizing retrieved documents substantially exceed gains from improving retrieval alone.

**Strengths:**

1. The motivation is clear and straightforward.

2. Experiments span six standard QA datasets, multiple retrievers (BM25, BGE, Contriever), and six mainstream 7–9B models, with analyses over top-k and retriever variants; the aggregate tables/figures make the pattern robust.

3. The paper reports how performance varies with number/quality of retrieved documents, documents containing answers, and retriever choice, highlighting when “better retrieval” stops paying off and why utilization matters more.

**Weaknesses:**

1. Limited novelty. The core conclusions, (i) even with near-oracle document quality, the marginal RAG gain is modest; (ii) better leveraging retrieved content in the generator is the higher-leverage path, are largely observational and align with prior RAG ablation/oracle-style studies. The two “oracle” approximations (query+answer retrieval; per-document prompting/voting) are close to ideas discussed in prior work including [1], so the methodological contribution feels incremental.

2. Presentation quality is weak. The method section is hard to follow and contains numerous grammatical issues and inconsistent notation, which obscures otherwise simple ideas. The paper should tighten the exposition: define symbols once, move toy figures into a compact schematic, and proofread carefully.

3. Related work is insufficient. A proper Related Work section is missing from the main paper; relegating material to the appendix cuts off essential context and still omits important lines of work (e.g., oracle/upper-bound analyses for RAG, selective/context routing and chunk-level voting, self-refinement/corrective RAG variants, retrieval-conditioned reasoning).

[1] Wang F, Wan X, Sun R, et al. Astute rag: Overcoming imperfect retrieval augmentation and knowledge conflicts for large language models[J]. arXiv preprint arXiv:2410.07176, 2024.

[2] Wang H, Li R, Jiang H, et al. Blendfilter: Advancing retrieval-augmented large language models via query generation blending and knowledge filtering[J]. arXiv preprint arXiv:2402.11129, 2024.

**Questions:**

refer to weaknesses

---

### Official Review · Reviewer_9dNq · 2025-11-01

**Soundness:** 2
**Presentation:** 2
**Contribution:** 2
**Rating:** 2
**Confidence:** 3

**Summary:**

This paper investigates the performance bottlenecks in traditional Retrieval-Augmented Generation (RAG) pipelines. Specifically, it analyzes both gold retrieval and gold utilization scenarios to pinpoint the sources of performance gaps. Experiments conducted on six question-answering datasets reveal that the primary bottleneck lies in the LLM’s inability to effectively leverage the retrieved documents. The paper also presents comprehensive ablation studies and analyses to further examine how performance is affected by both the quality of document utilization methods and the quality of the retrieved documents themselves.

**Strengths:**

- The motivation and central idea of investigating performance bottlenecks in traditional RAG systems are both strong and interesting. Indeed, most existing RAG approaches simply concatenate retrieved knowledge with the input in a shallow manner, which may prevent the model from fully leveraging the retrieved information and can introduce errors. This paper tries to investigate this issue by proposing two methods to quantify and compare performance gaps in the retrieval and generation stages.
- The paper conducts extensive experiments and analyses across six QA datasets, offering a clear and well-structured examination of performance differences between the two stages. The writing is generally clear, coherent, and easy to follow.

**Weaknesses:**

Overall, the paper’s contribution lacks sufficient novelty to be considered strong for a venue such as ICLR. Although extensive experiments are conducted on six datasets, the findings are relatively shallow and largely reaffirm conclusions already established in prior work. Moreover, the paper does not propose any new methods or solutions to address the identified performance gaps.

**Questions:**

Typo:
- L032, incorrect citation
- L151, L159, space

---

### Official Review · Reviewer_69Sv · 2025-11-01

**Soundness:** 2
**Presentation:** 3
**Contribution:** 2
**Rating:** 4
**Confidence:** 3

**Summary:**

The paper investigates the main bottleneck in LLM-based RAG systems by separately analyzing the retrieval and generation stages. The main finding is that enabling generation models to better utilize retrieved documents offers greater potential for improving RAG performance.

**Strengths:**

* The paper is well-structured.
* The proposed research questions are clear.
* The decoupled perspective is reasonable.
* The conclusions are clearly presented.

**Weaknesses:**

* The study is conducted on benchmarks primarily centered around Wikipedia. It would be valuable to explore the problem in more realistic RAG settings that go beyond Wikipedia and/or incorporate multiple sources.
* The evaluation is limited to models in the 7B–9B parameter range. It remains unclear whether the findings generalize to models of different sizes, particularly larger state-of-the-art models.
* The underlying idea that retrieval and generation are separate modules, and that one may be more limiting than the other (as well as analyses of where the bottlenecks lie) is not entirely novel.

**Questions:**

n/a

---

### Note · Authors · 2025-11-16

**Comment:**

Theme: Request for the withdrawal of paper

Dear Editor,​

First of all, we would like to express our sincere gratitude to you and the journal team for your attention and preliminary processing of our submitted paper (Title: Rethinking LLM-based RAG from a decoupled perspective; Submission Number: 18235).​
With deep regret, we are writing to formally request the withdrawal of our paper. Due to insufficient preparation in the early stage, the experimental analysis part of the paper was completed in a hurry, and there is still room for improvement in some data interpretation and argumentation logic. To ensure that the quality of the paper meets the publication standards of the journal and to present more rigorous and reliable research results to readers, we plan to comprehensively revise and polish the paper. At the same time, the subsequent improvement of the experiment requires additional time to carry out supplementary research, and some complex data analysis needs to allocate more computing resources for support. Therefore, we cannot meet the submission requirements at this stage.​
We are deeply sorry for the inconvenience caused to the journal's editorial work by this withdrawal and express our sincere apologies again. After the paper is revised and improved, we will re-submit the application in accordance with the journal's requirements and look forward to having the opportunity to receive your guidance and support again in the future.​
Thank you for your understanding and tolerance!​
Sincerely,​
Authors: Zhichun Xu, Conghui Zhu, Lemao Liu, Tiejun Zhao, Muyun Yang
November 16, 2025

**Withdrawal Confirmation:**

I have read and agree with the venue's withdrawal policy on behalf of myself and my co-authors.